# The implementation of prioritization exercises in the development and update of health practice guidelines: A scoping review

**Amena El-Harakeh**[1,2], **Tamara Lotfi**[3,4], **Ali Ahmad**[4], **Rami Z. Morsi**[4], **Racha Fadlallah**[1,5], **Lama Bou-Karroum**[1,5], **Elie A. Akl**[1,2,6,7]*

**1** Center for Systematic Reviews on Health Policy and Systems Research (SPARK), American University of Beirut, Beirut, Lebanon, **2** Clinical Research Institute (CRI), American University of Beirut Medical Center, Beirut, Lebanon, **3** Global Evidence Synthesis Initiative (GESI) Secretariat, American University of Beirut, Beirut, Lebanon, **4** Faculty of Medicine, American University of Beirut, Beirut, Lebanon, **5** Department of Health Management and Policy, Faculty of Health Sciences, American University of Beirut, Beirut, Lebanon, **6** Department of Internal Medicine, American University of Beirut Medical Center, Beirut, Lebanon, **7** Department of Health Research Methods, Evidence, and Impact (HEI), McMaster University, Hamilton, Canada

* ea32@aub.edu.lb

**Data Availability Statement:** All relevant data are within the paper and its Supporting Information files.

## Abstract

### Background

The development of trustworthy guidelines requires substantial investment of resources and time. This highlights the need to prioritize topics for guideline development and update.

### Objective

To systematically identify and describe prioritization exercises that have been conducted for the purpose of the de novo development, update or adaptation of health practice guidelines.

### Methods

We searched Medline and CINAHL electronic databases from inception to July 2019, supplemented by hand-searching Google Scholar and the reference lists of relevant studies. We included studies describing prioritization exercises that have been conducted during the de novo development, update or adaptation of guidelines addressing clinical, public health or health systems topics. Two reviewers worked independently and in duplicate to complete study selection and data extraction. We consolidated findings in a semi-quantitative and narrative way.

### Results

Out of 33,339 identified citations, twelve studies met the eligibility criteria. All included studies focused on prioritizing topics; none on questions or outcomes. While three exercises focused on updating guidelines, nine were on de novo development. All included studies addressed clinical topics. We adopted a framework that categorizes prioritization into 11 steps clustered in three phases (pre-prioritization, prioritization and post-prioritization). Four studies covered more than half of the 11 prioritization steps across the three phases. The

**Funding:** This study was supported by the Alliance for Health Policy and Systems Research, World Health Organization (WHO), Geneva. The funder was not involved in the design of the study, in the collection, analysis, and interpretation of data or in the writing of the manuscript.

most frequently reported steps for generating initial list of topics were stakeholders' input (n = 8) and literature review (n = 7). The application of criteria to determine research priorities was used in eight studies. We used and updated a common framework of 22 prioritization criteria, clustered in 6 domains. The most frequently reported criteria related to the health burden of disease (n = 9) and potential impact of the intervention on health outcomes (n = 5). All the studies involved health care providers in the prioritization exercises. Only one study involved patients. There was a variation in the number and type of the prioritization exercises' outputs.

## Conclusions

This review included 12 prioritization exercises that addressed different aspects of priority setting for guideline development and update that can guide the work of researchers, funders, and other stakeholders seeking to prioritize guideline topics.

## Introduction

Health practice guidelines are "systematically developed statements" intended to optimize care at the clinical, public health and health systems levels [1]. The development of trustworthy guidelines requires substantial investment of resources [2, 3] and time, often taking an average of two to three years [4, 5]. In addition, and with the expansion of medical research and the emergence of new and innovative technologies, guidelines should be updated as necessary [6]. This highlights the need to prioritize topics for guideline development and update.

In fact, the importance of prioritization for guideline development has been recognized by many guideline developing groups [7]. It ensures that limited resources are aligned with priority needs for guideline development [8]. Furthermore, prioritization exercises engaging a wide range of stakeholders enhance the relevance and potential uptake of priority topics by end users [9–11]. This represents an important step toward improving the delivery of evidence-informed care.

The guideline development process includes, in addition to prioritizing topics, the prioritization of questions and outcomes [5, 12]. Also, prioritization should be considered when adapting guidelines to select priority questions from among those addressed in the original guidelines [13]. Similarly, guideline developers need to prioritize which guidelines, guideline sections, or recommendations should be updated [14].

With the growing interest among researchers in prioritizing topics for the de novo development, update and adaptation of guidelines, several exercises have been conducted to yield explicit and transparent prioritization [15–17]. Some investigators relied primarily on the use of criteria to select priority guideline topics [18], while others have followed multicomponent prioritization processes and have used established tools and approaches [19].

While some efforts have been invested in synthesizing the evidence on prioritization for guideline updating [20], none have described prioritization for the de novo development, update or adaptation of guidelines. As such, the objective of this study was to systematically identify and describe prioritization exercises that have been conducted for the purpose of the de novo development, update or adaptation of health practice guidelines.

## Methods

We conducted a scoping review of published prioritization exercises implemented as part of the de novo development, update or adaptation of health practice guidelines. We followed

standard methodology and the Preferred Reporting Items for Systematic Reviews and Meta-Analyses extension for Scoping Reviews (PRISMA-ScR) guidelines for reporting scoping reviews [21] (see S1 File). This study is based on a detailed protocol available in S2 File. The project's team included a multidisciplinary group of professionals in the clinical, public health, and health policy and systems fields, with expertise in guideline development and priority setting.

### Eligibility criteria

- Paper type: We included descriptive reports and excluded commentaries, editorials, letters, correspondences, news, and abstracts.

- Scope: We included papers about the de novo development, update or adaptation of health practice guidelines addressing clinical, public health or health system topics. Also, we excluded papers reporting proposed approaches, without any applied exercise.

- Focus: We included papers that aimed to prioritize one of the following: guideline topics, questions/recommendations, or outcomes. We excluded papers reporting on individual prioritization criteria or items.

- Setting: We included eligible papers irrespective of the setting (low-, middle- or high-income countries; primary, secondary or tertiary healthcare facilities).

### Information sources and literature search

We searched Medline and CINAHL electronic databases from their respective inception date to July 2019. We also manually searched Google Scholar in July 2019. We developed the search strategy with the assistance of a medical librarian. The search included both Medical Subject Headings (MeSH) terms and free-text words and combined various terms for health prioritization (see S3 File). We did not use any language or date restrictions. We screened the reference lists of included and other relevant papers.

### Study selection

The study selection process consisted of two phases: title and abstract screening and full text screening. Teams of two reviewers (AEH, TL, AA, RZM, RF, LBK) independently and in duplicate screened the titles and abstracts of citations captured by the search for potential eligibility. The reviewers then obtained the full texts of citations judged as potentially eligible by at least one of the two reviewers. Then, they screened the full texts in duplicate and independently for eligibility using a standardized and pilot-tested screening form and following a calibration exercise. At this phase, the reviewers resolved disagreements by discussion or with the input of a third reviewer (EAA) when consensus could not be reached.

### Data extraction

Prior to data extraction, we conducted two calibration exercises to enhance the validity of the process. Three reviewers (AEH, TL and AA) worked in duplicate and independently using a standardized and pilot-tested data extraction form (see S4 File). They resolved disagreements by discussion or with the help of a third reviewer (EAA). We collected the following data from each included paper:

- General characteristics: authors, lead entity, target audience, year of prioritization conduct, scope of prioritization, topic (specific domain), focus of prioritization, type of guideline development, and support/funding;

- Steps of prioritization:

  - Pre-prioritization phase (development of guiding/ethical principles, generation of initial list of topics and collection of technical data to inform discussions);

  - Prioritization phase (use of established prioritization methods, research gap analysis, use of criteria, prioritization/ranking);

  - Post-prioritization phase (refinement of priorities into guideline topics, dissemination and implementation, revision mechanism, monitoring and evaluation).

### Data synthesis

Due to the descriptive nature of data, we synthesized the findings in a semi-quantitative way. We used the extracted data to come up with common categorizations of relevant concepts (e.g., prioritization steps, generation of initial list of topics), using an iterative process of review and refinement. As part of this process, we analyzed the content of each study at least twice; when drafting the initial categories, and after producing an advanced draft. We reported the results narratively and in a tabular format.

The concepts addressed in our analysis included:

- Prioritization steps; we adopted 11 categories of prioritization steps, which we developed for a recent systematic review on prioritization for evidence synthesis [22];

- Generation of initial list of topics (descriptive analysis);

- Output of the priority setting exercises (descriptive analysis);

- Stakeholder involvement; we adopted the categories we developed for a recent systematic review on prioritization for evidence synthesis [22], which is based on the 7Ps framework [23];

- Prioritization criteria; we used a common framework of prioritization criteria that we developed for a recent systematic review on prioritization approaches in the development of health practice guidelines [24] (see S5 File).

## Results

### Study selection

Fig 1 shows the study flow diagram summarizing the study selection process. Out of the 33,339 identified citations, twelve papers met our inclusion criteria [14–19, 25–30]. We excluded 896 articles based on full text screening for the following reasons: not a paper type of interest (n = 49); not describing a reproducible prioritization exercise (n = 322); not about health practice guidelines (n = 525).

### General characteristics of the included studies

Table 1 shows the general characteristics of the twelve included studies. One prioritization exercise was conducted in 1998 while the remaining ones were conducted between 2010 and 2017. Half of the prioritization exercises were implemented at a national level (n = 6) [14, 16, 19, 27, 29, 30], while the rest were implemented at regional (n = 3) [25, 26, 28], provincial

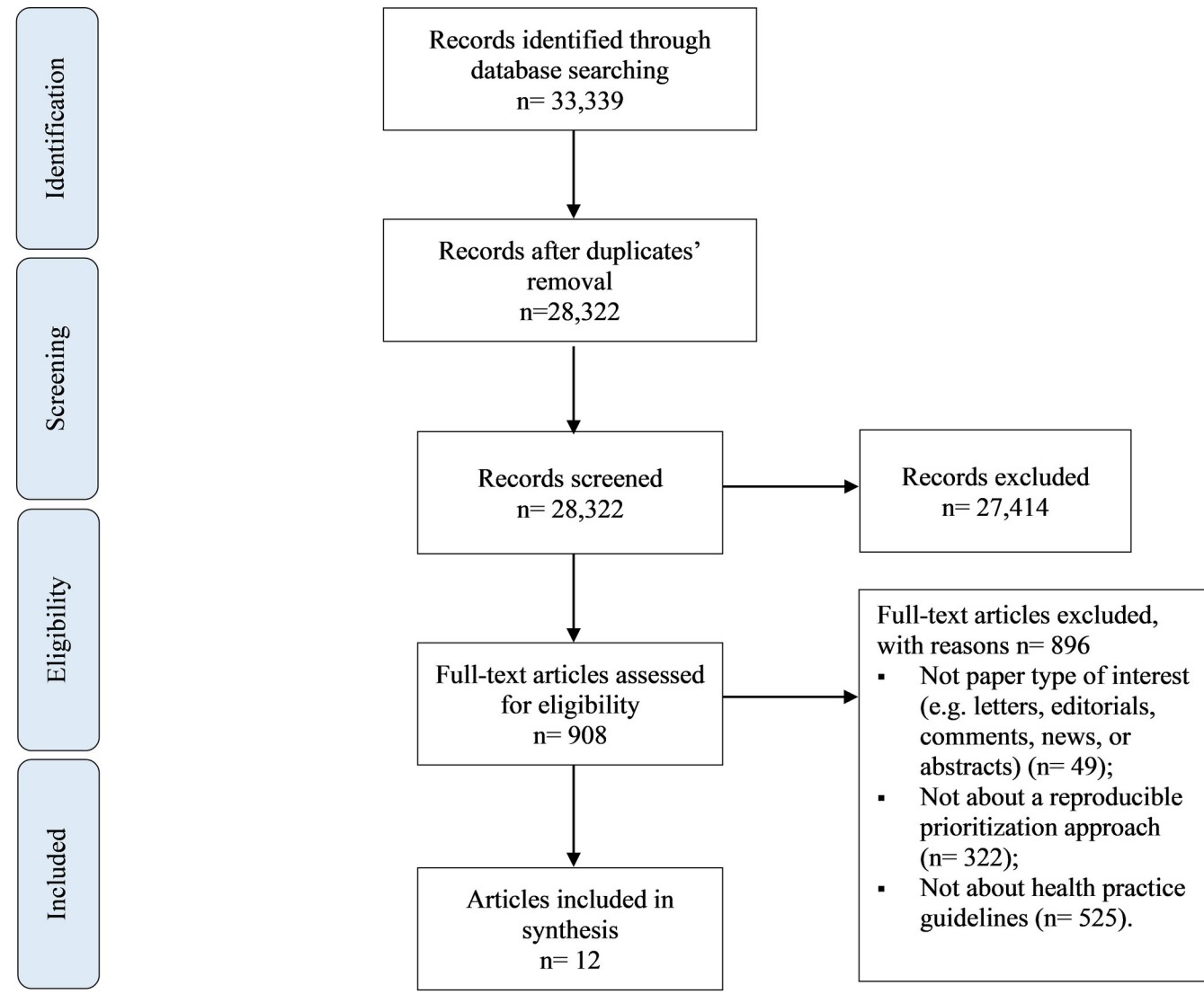

**Fig 1. Study flow diagram.**

(n = 2) [15, 18], or international levels (n = 1) [17]. All of the prioritization exercises focused on prioritizing guideline topics (as opposed to prioritizing questions or outcomes) and addressed clinical topics. While three exercises focused on updating guidelines [14, 15, 26], nine focused on de novo development [16–19, 25, 27–30], and none addressed adaptation of guidelines. Funding sources were mainly professional societies (n = 5) [16, 25, 26, 28, 29] or public funding (n = 4) [15, 18, 19, 30].

We present below our findings summarized as the steps of prioritization, with a focus on two of those steps for which detailed information was available (namely the generation of initial list of topics and prioritization criteria). Then, we review stakeholder involvement in the prioritization exercises. Finally, we review the processes and outputs of the prioritization exercises stratified by whether the prioritization exercise was related to de novo development or updating.

**Table 1. General characteristics of the exercises for prioritizing guideline topics.**

| Study | Lead entity | Target audience | Year of prioritization conduct | Scope of prioritization | Topic (specific domain) | Focus of prioritization | Type of guideline development | Support/ Funding |
|---|---|---|---|---|---|---|---|---|
| Agbassi, 2014 [15] | Program in Evidence-based Care, Clinical Practice Guideline Initiative of the Ontario provincial cancer system | Clinical Practice Guidelines developing groups | 2011–2012 | Provincial (Ontario) | Clinical (cancer care) | Topics | Update | Funded by the Cancer Care Ontario |
| Brouwers, 2003 [18] | Practice Guidelines Initiative of Cancer Care Ontario's Program in Evidence-based Care | Clinicians | Not reported[1] | Provincial (Ontario) | Clinical (role of prophylactic anticonvulsant drugs in brain tumors) | Topics | De novo | "Sponsored by Cancer Care Ontario and Ontario Ministry of Health and Long-Term Care" |
| Borgonjen, 2015 [16] | Not reported | Dermatological professional organizations | 2010 | National (Netherlands) | Clinical (dermatological diseases) | Topics | De novo | "Supported by Dutch Society of Dermatology and Venereology" |
| Becker, 2018 [14] | Not reported | Clinical Practice Guidelines groups | 2014–2015 | National (Germany) | Clinical (acute perioperative and posttraumatic pain) | Topic (Guideline sections) | Update[2] | Not funded |
| Farrell, 2015 [30] | Not reported | Clinicians | 2013–2014 | National (Canada) | Clinical (medication/drug classes for deprescription) | Topics | De novo | Funded by the Ontario Ministry of Health and Long-Term Care |
| Jo, 2015 [19] | Korean Academy of Medical Sciences (KAMS) and Korea Centers for Disease Control and Prevention (KCDC) | Clinical Practice Guidelines developing groups (KAMS and KCDC) | 2013–2014 | National (Korea) | Clinical (chronic diseases) | Topics | De novo | "Supported by Korea Health Promotion Foundation and Korea Centers for Disease Control and Prevention" |
| Kerr, 2009 [17] | Not reported | Clinicians (epilepsy specialists) | Not reported | International (6 countries; not specified) | Clinical (epilepsy in adults with intellectual disability) | Topics | De novo | Not reported |
| Loeffen, 2015 [29] | Not reported | International Clinical Practice Guidelines developers | 2013 | National (Netherlands) | Clinical (supportive care in pediatric cancer) | Topics | De novo | "Supported by Dutch Cancer Society" |
| Nast, 2019 [28] | European Academy of Dermatology and Venereology and the Division of Evidence-Based Medicine, Charité–Universitätsmedizin Berlin | Clinicians (dermatologists) | 2017 | Regional (Europe) | Clinical (dermatology and venereology) | Topics | De novo | Funded by the European Academy of Dermatology and Venereology |
| van der Sanden, 2002 [27] | Not reported | Dutch Dental Association | 1998 | National (Netherlands) | Clinical (dentistry) | Topics | De novo | Not reported |

(*Continued*)

**Table 1.** (Continued)

| Study | Lead entity | Target audience | Year of prioritization conduct | Scope of prioritization | Topic (specific domain) | Focus of prioritization | Type of guideline development | Support/ Funding |
|-------|-------------|-----------------|-------------------------------|------------------------|------------------------|------------------------|------------------------------|------------------|
| van der Veer, 2015 [26] | European Renal Best Practice | Clinical Practice Guidelines developing groups | 2014 | Regional (Europe) | Clinical (vascular access for haemodialysis) | Topics | Update | "Supported by a grant of the European Renal Association-European Dialysis Transplant Association" |
| van der Veer, 2016 [25] | European Renal Best Practice | Clinicians (Nephrologists and geriatricians) | Not reported | Regional (Europe) | Clinical (chronic kidney disease in older adults) | Topics | De novo | "Financially endorsed by the European Renal Association-European Dialysis Transplant Association" |

[1] The projected completion date of the guideline was Winter, 2004.

[2] A priori, the guideline steering group decided not to update the whole guideline.

## Steps of prioritization

Table 2 outlines the prioritization steps addressed in the twelve included studies across three phases: pre-prioritization, prioritization and post-prioritization phases. Although most of the studies (n = 11) addressed at least one step in the pre-prioritization phase [14–17, 19, 25–30], less than half (n = 5) addressed at least one step during the post-prioritization phase [14, 15, 17, 25, 27]. Four studies covered more than half of the 11 prioritization steps across the three phases [14, 15, 19, 25].

Prior to conducting the prioritization exercises, most of the included studies (n = 10) generated initial lists of topics [15–17, 19, 25–30], while only a few studies reported on the development of ethical principles to guide the conduct of the exercise (n = 3) [25, 26, 30], or on the collection of technical data to inform further discussions (n = 2) [14, 19].

Most studies used prioritization criteria (n = 8) [14, 15, 18, 19, 25, 26, 29, 30] and ranked priorities (n = 11) [14–17, 19, 25–30] during the prioritization phase. Out of eight studies, two refined priorities into guideline topics (excluding four studies where this step was not applicable as the exercise started with topics and not broad themes) [17, 25]. Less than half of the studies conducted or reported on a plan for dissemination and implementation (n = 3) [15, 17, 25] or monitoring and evaluation (n = 3) [14, 15, 27]. All of the studies involved stakeholders during various prioritization steps across the three phases, with the majority involving stakeholders in the generation of initial list of topics (n = 8) [16, 17, 25–30], use of criteria (n = 8) [14, 15, 18, 19, 25, 26, 29, 30] and prioritization/ranking of priorities (n = 10) [14–17, 19, 25, 26, 28–30]. Only one study engaged stakeholders in the post-prioritization phase [17].

**Generation of initial list of topics.** Table 3 shows the steps involved in generating initial list of topics. One frequently used method for generating initial list of topics was seeking input from stakeholders (n = 8) [16, 17, 25–30]. Other methods included reviewing the literature (n = 7) [15, 17, 19, 25–27, 30], referring to the health information system (n = 1) [19] and to previous priority setting exercises (n = 1) [15].

**Prioritization criteria.** Table 4 presents the prioritization criteria that 10 out of the 12 studies reported on. Eight studies used their proposed criteria as part of their prioritization

**Table 2.** Steps addressed when prioritizing guideline topics [1].

| Paper | Pre-prioritization phase | | | | Prioritization phase | | | Post-prioritization phase | | | |
|---|---|---|---|---|---|---|---|---|---|---|---|
| | Development of guiding/ethical principles | Generation of initial list of topics | Collection of technical data to inform discussions | Use of established prioritization methods | Research gap analysis | Use of prioritization criteria | Prioritization/ Ranking | Refinement of priorities into guideline topics | Dissemination and implementation of priorities | Revision mechanism | Monitoring and evaluation |
| % of studies | n = 3 25% | n = 10 83% | n = 2 17% | n = 3 25% | n = 2 17% | n = 8 67% | n = 11 92% | n = 2 17% | n = 3 25% | n = 0 0% | n = 3 25% |
| **De novo development of guidelines** | | | | | | | | | | | |
| Brouwers, 2003 [18] | | | | | | | | N/A | | | |
| Borgonjen, 2015 [16] | | ✓[2] | | | | | ✓ | | | | |
| Farrell, 2015 [30] | ✓ | ✓ | ✓ | | | ✓ | ✓ | N/A | | | |
| Jo, 2015 [19] | | ✓ | ✓ | ✓ | ✓ | ✓ | ✓ | | | | |
| Kerr, 2009 [17] | | ✓ | | | | ✓ | ✓ | ✓ | ✓ | | |
| Loeffen, 2015 [29] | | ✓ | | | | ✓ | ✓ | | | | |
| Nast, 2019 [28] | | ✓ | | | ✓ | | ✓ | | | | |
| van der Sanden, 2002 [27] | | | | | | | ✓ | | | | ✓ |
| van der Veer, 2016 [25] | ✓ | ✓ | | | | ✓ | ✓ | ✓ | ✓ | | |
| **Updating of guidelines** | | | | | | | | | | | |
| Agbassi, 2014 [15] | | ✓ | | ✓[3] | N/A | ✓ | ✓ | N/A | ✓ | | ✓ |
| Becker, 2018 [14] | | | ✓ | ✓[4] | N/A | ✓ | ✓ | N/A | | | ✓ |
| van der Veer, 2015 [26] | ✓ | ✓ | | | N/A | ✓ | ✓ | | | | |

[1] We did not include 'stakeholder engagement' as a separate step given that it was included across the three phases. The colored cells denote that stakeholders were engaged in the step.

[2] Unclear whether participants were given an initial list and asked to suggest additional ones, or if it was based on suggestions only.

[3] The exercise is based on former updating procedures of the Program in Evidence-based Care and procedures of other established guideline developing groups (e.g., Scottish Intercollegiate Guidelines Network and National Institute for Health and Care Excellence).

[4] The pilot study followed a previously developed updating procedure (using information from a systematic review on the methods for updating clinical practice guidelines).

**Table 3. Steps involved in generating initial list of topics.**

| Paper | Literature review | | | | Health information system | Previous priority setting exercises | Stakeholder input | Refinement of the initial list of topics |
|---|---|---|---|---|---|---|---|---|
| | Existing trial | Existing systematic review | Existing practice guideline | Other | | | | |
| **% of studies** | n = 3 25% | n = 2 17% | n = 4 33% | n = 3 25% | n = 1 8% | n = 1 8% | n = 8 67% | n = 4 33% |
| **De novo development of guidelines** | | | | | | | | |
| Brouwers, 2003[1] [18] | | | | | | | | |
| Borgonjen, 2015 [16] | | | | | | | ✓ | |
| Farrell, 2015 [30] | | | | ✓[2] | | | ✓ | |
| Jo, 2015 [19] | | | ✓ | ✓[3] | ✓ | | | |
| Kerr, 2009 [17] | ✓ | ✓ | | | | | ✓ | |
| Loeffen, 2015 [29] | | | | | | | ✓ | |
| Nast, 2019 [28] | | | | | | | ✓ | ✓ |
| van der Sanden, 2002 [27] | | | | ✓[4] | | | ✓ | ✓ |
| van der Veer, 2016 [25] | ✓ | | ✓ | | | | ✓ | ✓ |
| **Updating of guidelines** | | | | | | | | |
| Agbassi, 2014 [15] | | | ✓ | | | ✓ | | |
| Becker, 2018 [14] | | | | | | | | |
| van der Veer, 2015 [26] | ✓ | ✓ | ✓ | | | | ✓ | ✓ |

[1] The guideline topic was identified by the Neuro-oncology disease site group and was then confirmed through surveying practitioners.

[2] The type of literature searched was not specified (i.e., reports outlining the prevalence and impact of polypharmacy).

[3] Global Burden of Disease (GBD).

[4] Original contributions, clinical reports, editorials and letters to the editor (1992–1997), and analysis of discussions of dental peer groups (1989–1998).

exercises [14, 15, 18, 19, 25, 26, 29, 30], while two studies proposed criteria but did not use them in the exercise [16, 28].

The studies included a mean of seven prioritization criteria (range: 3–13), with a total of 70 criteria reported. We attempted to match the 70 criteria to a published framework of 20 guideline prioritization criteria classified into six domains (Table 4 and S5 File). During the matching process, we added two criteria that emerged from the included studies (i.e., availability of low certainty evidence and acceptability). Table 4 shows the classification of the identified prioritization criteria according to the new framework.

**Table 4. Prioritization criteria[1] and the domains they fall under (n = 10).**

| Paper | Disease-related factors | | | | | Interest | | | Practice | | Guideline development | | | | | Potential impact of the intervention | | | | Implementation considerations | | |
|---|---|---|---|---|---|---|---|---|---|---|---|---|---|---|---|---|---|---|---|---|---|---|
| **% papers reporting the criterion** | Health burden | Economic burden | Burden on healthcare system | Equity relevance | Urgency | Health professional level | Consumer level | National level | Practice variation | Uncertainty or controversy about best practice | Absence of guidance | Unsatisfactory guidance | Availability of evidence | Availability of low certainty evidence | Potential for changing existing guidance | Impact on health outcomes | Economic impact | Impact on the healthcare system | Impact on equity/ access | Feasibility of intervention implementation | Availability of resources | Acceptability |
|  | n = 9 90% | n = 3 30% | n = 3 30% | n = 0 0% | n = 0 0% | n = 3 30% | n = 3 30% | n = 1 10% | n = 3 30% | n = 3 30% | n = 2 20% | n = 2 20% | n = 3 30% | n = 2 20% | n = 3 30% | n = 5 50% | n = 2 20% | n = 2 20% | n = 1 10% | n = 2 20% | n = 1 10% | n = 1 10% |
| *De novo development of guidelines* | | | | | | | | | | | | | | | | | | | | | | |
| Brouwers, 2003 [18] | ✓ | | | | | ✓ | | | | | | | | | | | | | | | | |
| Borgonjen, 2015[2] [16] | ✓ | | | | | ✓ | ✓ | ✓ | ✓ | | | | ✓ | ✓ | | ✓ | ✓ | ✓ | | ✓ | | |
| Farrell, 2015 [30] | | | | | | | ✓ | | | | ✓ | | ✓ | ✓ | | ✓ | ✓ | ✓ | | ✓ | | ✓ |
| Jo, 2015 [19] | ✓ | ✓ | | | | | | | | | | | | | | | | | | | | |
| Loeffen, 2015 [29] | ✓ | | | | | | | | | | | | | | | ✓ | | | | | | |
| Nast, 2019[2] [28] | ✓ | ✓ | ✓ | | | | | | ✓ | | | | | | ✓ | | | | | | | |
| van der Veer, 2016 [25] | ✓ | | ✓ | | | | ✓ | | | ✓ | | | | | | ✓ | | | | | | |
| *Updating of guidelines* | | | | | | | | | | | | | | | | | | | | | | |
| Agbassi, 2014 [15] | ✓ | | | | | | | | | | | ✓ | ✓ | | ✓ | | | | ✓ | | | |
| Becker, 2018 [14] | ✓ | | | | | | | | ✓ | ✓ | ✓ | ✓ | | | ✓ | | | | | | ✓ | |
| van der Veer, 2015 [26] | ✓ | ✓ | ✓ | | | ✓ | | | | ✓ | | | | | | ✓ | | | | | | |

[1] All prioritization criteria have been worded in a way favoring prioritization (i.e., a favorable assessment of the criterion indicates higher priority).

[2] Criteria were proposed but were not used in the prioritization exercise.

The most frequently reported criteria related to the health burden of disease (n = 9) [14–16, 18, 19, 25, 26, 28, 29] and potential impact of the intervention on health outcomes (n = 5) [16, 25, 26, 29, 30]. None of the studies included equity relevance of the disease or urgency as explicit criteria. Eleven (out of the total of 22 criteria listed in the framework) was the highest number of criteria reported by a study [16].

## Stakeholder involvement

Table 5 shows the types of stakeholders involved in prioritizing guideline topics and the methods used to engage them. All included studies involved healthcare providers in the prioritization exercises, while four studies (33%) involved researchers [14, 19, 29, 30], and only one study (8%) involved patients (Table 5) [26]. In Loeffen et al., the authors reported not including patients, parents, and caretakers as they planned to understand the needs of the professionals beforehand [29]. Seven studies described stakeholder recruitment methods which ranged from the use of professional networks (e.g., members directory), to searching databases and emailing clinicians [16–18, 25, 26, 28, 30].

All prioritization exercises surveyed stakeholders (e.g., Delphi approach) as a method of engagement. Other methods included the nominal group technique (n = 1) [25] and consensus conference (n = 1) [14]. Stakeholders were engaged via an online platform (e.g., online surveys, email discussions) in all included studies, with two studies using both online and in-person meetings [14, 25]. The frequency of engagement varied from only once (n = 4), to twice (n = 6), or three times (n = 2).

## Prioritization processes and outputs for de novo development (n = 9)

Table 6 describes the processes and outputs of the prioritization exercises. The nine studies that implemented prioritization processes for the de novo development of guidelines followed common steps of reviewing the literature (n = 5) [17, 19, 25, 27, 30] and/or engaging stakeholders (n = 9) [16–19, 25, 27–30], while considering the availability of existing guidelines on the suggested topics (n = 3) [16, 19, 28]. In fact, one study conducted the prioritization exercise regardless of existing guidelines which resulted in prioritizing 20 topics; all of which were covered by existing guidelines [16].

There was a variation in the types of outputs of the prioritization exercises. Most of the studies prioritized topics (n = 6) [16, 18, 25, 27–29], one prioritized clinical areas (n = 1) [17], one prioritized drug classes (n = 1) [30], and one prioritized chronic diseases (n = 1) [19]. None prioritized questions or outcomes. Most of the studies provided ranked lists of priorities (n = 8) [16, 17, 19, 25, 27–30], while one study had a topic suggested prior to the exercise and then confirmed as a result of prioritization [18]. The numbers of priorities derived from the initial lists varied between the studies (range 1–46).

## Prioritization processes and outputs for updating (n = 3)

Studies that implemented prioritization exercises for updating (n = 3) either assessed candidate guideline documents for updating [15] or selected a specific guideline a priori and assessed potential topics or sections to be covered by updating [14, 26].

Agbassi et al. used a stepwise process in which two questionnaires were implemented to prioritize guidelines for updating and to assess the effect of new evidence on existing recommendations [15]. van der Veer et al. consulted clinicians and patients about priority topics to be covered by the update of 2007 vascular access guideline of the European Renal Best Practice [26]. Becker et al. classified guideline sections of a German clinical practice guideline based on evidence and clinical relevance (Table 6) [14]. Two studies used categories (e.g., urgent, high,

**Table 5.  Types of stakeholders involved in prioritizing guideline topics and the methods of engagement.**

| Paper | Types of stakeholders | | | | | | | | | | | | | Description of recruitment method | Method(s) of engagement |
|---|---|---|---|---|---|---|---|---|---|---|---|---|---|---|---|
| | Public policymakers | Health care providers | Researchers | Members of the public | Patients and their representatives | Caregivers | Health system payers | Health care managers | Intergovernmental agencies/Research funders | Product makers/Industry | Press & journalists | Non-governmental organizations | Other | | |
| % of studies reporting | n = 0 0% | n = 12 100% | n = 4 33% | n = 0 0% | n = 1 8% | n = 0 0% | n = 0 0% | | n = 0 0% | n = 0 0% | n = 0 0% | n = 0 0% | n = 0 0% | n = 7 58% | |
| *De novo development of guidelines* | | | | | | | | | | | | | | | |
| Brouwers, 2003 [18] | | ✓ | | | | | | | | | | | | ✓ | **Prioritization:** modified Dillman (online survey via email) |
| Borgonjen, 2015 [16] | | ✓ | | | | | | | | | | | | ✓ | **Generation and Prioritization:** online survey |
| Farrell, 2015 [30] | | ✓ | ✓ | | | | | | | | | | | ✓ | **Generation:** modified Delphi (round 1 of online survey via email) **Prioritization:** modified Delphi (rounds 2 and 3 online survey via email) |
| Jo, 2015 [19] | | ✓ | ✓ | | | | | | | | | | | | **Prioritization:** online survey via email |
| Kerr, 2009 [17] | | ✓ | | | | | | | | | | | | ✓ | **Generation:** modified Delphi (round 1 of online survey via email) **Prioritization:** modified Delphi (round 2 of online survey via email) |
| Loeffen, 2015 [29] | | ✓ | ✓ | | | | | | | | | | | | **Generation:** Delphi (round 1 of online survey via email) **Prioritization:** Delphi (round 2 of online survey vi email) |
| Nast, 2019 [28] | | ✓ | | | | | | | | | | | | ✓ | **Generation:** online piloted survey via email (round 1) **Prioritization:** online piloted survey via email (round 2) |
| van der Sanden, 2002 [27] | | ✓ | | | | | | | | | | | | | **Generation:** online survey |
| van der Veer, 2016 [25] | | ✓ | | | | | | | | | | | | ✓ | **Generation:** open consultation (online piloted survey) **Prioritization:** open consultation (online piloted survey) & expert consensus meeting (nominal group technique: 2-round voting) |
| *Updating of guidelines* | | | | | | | | | | | | | | | |
| Aghassi, 2014[1] [15] | | ✓ | | | | | | | | | | | | | **Prioritization:** completion of 2 surveys: annual document assessment and document review in 2 iterations (2 assessment and review cycles) |
| Becker, 2018 [14] | | ✓ | ✓ | | | | | | | | | | | | **Prioritization:** online survey via email and consensus conference |
| van der Veer, 2015 [26] | | ✓ | | | ✓ | | | | | | | | | ✓ | **Generation:** online piloted survey (round 1) **Prioritization:** online piloted survey (rounds 1 and 2) |

[1] Following the initiation of a review, review outcomes (endorsement, archive, and update) should be approved by a larger expert panel comprising a multidisciplinary team of clinicians and other stakeholders.

**Table 6. Prioritization processes and outputs of the conducted prioritization exercises.**

| Study | Initial list of priorities | Process (steps starting with initial list and ending with final list of priorities) | Output (final list of priorities) |
|---|---|---|---|
| **De novo development of guidelines** | | | |
| Brouwers, 2003 [18] | One suggested topic | → One topic was identified by the Neuro-oncology disease site group (DSG)<br>→ Members of the DSG conducted a survey of Ontario clinicians to confirm topic selection and determine their support for the topic<br>→ Survey results showed variation in current practice and that practitioners feel that a guideline is necessary<br>→ One topic was confirmed for guideline development | One confirmed topic |
| Borgonjen, 2015 [16] | 157 topics and 10 criteria | → 157 dermatological topics were selected and ranked by 118 dermatologists as priority topics regardless of existing guidelines via a survey<br>→ 10 criteria were included for ranking by dermatologists who were asked to add any missing criterion<br>→ 20 topics were prioritized based on pooled scores; all of which were covered by existing guidelines; with an overlap between topics 12–20 and a further set of 15 topics and a significant difference in raking between top 5 and other 30 topics.<br>→ 8 criteria were prioritized, and 3 additional criteria were suggested | Ranked list of 20 topics with additional 15 topics (overlapping confidence intervals) Ranked list of 8 criteria |
| Farrell, 2015 [30] | 29 drug/drug classes | → 29 drug/drug classes were identified by research team and included in round one of survey; with 14 drug/drug classes reaching required consensus level (>70%) and 2 new drug classes added from comments<br>→ In round two, participants were asked to rank the 2 new drug classes, and were asked to re-rank the 14 drugs/drug classes while considering the overall round one results and justifying selection of the top 5 choices<br>→ In round 3, participants were asked to rank the top 5 drug classes using specific criteria<br>→ Top 14 drug classes were identified using the overall mean rank and by profession | Ranked list of 14 drug classes |
| Jo, 2015 [19] | 41 chronic diseases | → 41 chronic diseases were selected based on:<br> • Global Burden of Disease, 2012 Health Insurance Statistics Yearbook and ICD-10<br> • excluding diseases for which guidelines are being developed by KAMS<br>→ Data on burden of disease and a list of available guidelines were provided to the experts for consideration in prioritization<br>→ Analytic hierarchy process and subjective assessment were used for ranking<br>→ Top 20 diseases were derived from each assessment | Two ranked lists of 20 chronic diseases |
| Kerr, 2009 [17] | 30 clinical areas | → An initial list of 30 clinical areas was derived based on the below steps:<br> • experts were sent a CD-ROM with detailed evidence from Cochrane review (abstracts from review and all interventional trials identified)<br> • each expert was invited to submit 10 areas where "it was felt new clinical guidance statements were necessary"<br>→ Participants were then asked to rank the top 10 areas in another mailing<br>→ A large difference was observed in ranking score after area ranked 11 so this was the cut-off point | Ranked list of 11 clinical areas |
| Loeffen, 2015 [29] | 41 topics | → 41 topics were suggested by core team then rated by experts in round one of survey<br>→ 10 additional topics were suggested by experts<br>→ 21 topics were excluded from round two (16 topics had mean scores < 2.5 on one of the 3 criteria and 5 topics were not in top 20)<br>→ 30 topics (top 20 topics and the 10 additions) were sent to experts for further rating<br>→ Top 10 topics were identified using the overall mean rank and by profession | Ranked list of 10 topics |

*(Continued)*

**Table 6.** (Continued)

| Study | Initial list of priorities | Process (steps starting with initial list and ending with final list of priorities) | Output (final list of priorities) |
|---|---|---|---|
| Nast, 2019 [28] | 265 topics under disease categories | → 265 topics were suggested by participants of round 1 of the survey<br>→ Suggestions were combined into 35 broader topics to be ranked by participants<br>→ Suggestions well covered by existing guidelines or not specific to decide whether they are covered by existing guidelines were excluded<br>→ Respondents were asked to rank their top 10 topics in round 2 of the survey<br>→ Top 10 topics were generated according to total weighted points | Ranked list of 10 topics |
| van der Sanden, 2002 [27] | 1027 topics | → 1027 topics were obtained from 3 methods (survey, peer group, and literature) as follow:<br>  • Survey: dentists confirming important of guideline development proposed a maximum of 5 topics with justification<br>  • Analysis of peer group (8–10 dental practitioners) discussions over 1989–1998<br>  • All national dental journals and periodicals (n = 8) were screened over 1992–1997<br>→ 4th method: linear regression lines of topics obtained from literature<br>→ Two researchers collected data and reclassified topics into 9 groups; topics reported less than 6 times within a method were excluded<br>→ An overall rank was obtained by adding rank numbers assigned to the 4 methods for each topic; lowest value indicated highest priority<br>→ The reliability of a method was tested by the "item-rest sum correlation"; a correlation of rank positions of one method with the sum of rank positions obtained by the other 3 methods<br>→ 34 topics were prioritized as a result of ranking and reclassification | Ranked list of 34 topics belonging to 9 topic groups |
| van der Veer, 2016 [25] | 48 topics in 6 categories | → 48 topics in 6 categories were generated from a scoping literature review (813 titles) and views of an international expert panel<br>→ List was refined by the panel into 47 topics in 7 categories via an open online consultation of professionals (asked to select 50% of topics within each category as priority topics and add suggestions)<br>→ Expert consensus meeting with 2-round voting yielded a ranked list of the 46 topics based on overall median and range. One topic was not included in the rating process due to an administrative error | Ranked list of 46 topics in 7 categories |
| **Updating of guidelines** | | | |
| Agbassi, 2014 [15] | 151 PEBC guideline documents | → 151 PEBC guideline documents were assessed in consultation with a clinical expert and facilitated by a methodologist: 37 archived, 33 deferred, 6 special cases and 75 need review<br>→ Documents within review category underwent prioritization into low- (n = 20), medium- (n = 10), high- (n = 18) and urgent-priority (n = 27)<br>→ In order of priority, clinical expert and methodologist reviewed guidelines (via conducting a streamlined systematic review) to determine effect of new evidence on existing recommendation and further action<br>→ Then documents were classified as either endorsed (n = 15), updated (n = 8) or archived (n = 7), while others either required a new version (n = 1), had a review initiated (n = 7) or the process was incomplete (n = 35)<br>→ Review outcomes (endorsement, archive, and update) should be approved by a larger expert panel comprising a multidisciplinary team of clinicians and other stakeholders | Categorization of guideline documents into endorse (n = 15), update (n = 8) or archive (n = 7) |
| Becker, 2018 [14] | 35 guideline sections | → Limited search yielded 902 abstracts of potentially relevant evidence on the 35 guideline sections<br>→ Further new evidence was identified via an online survey of CGP group members who also rated the sections based on evidence and clinical relevance<br>→ Sections were subdivided in groups with high (15), medium (9), or low (11) need for update based on median scores<br>→ A consensus conference was held to finalize the list of sections with "high" need for update; 7 sections were allocated from low and middle to high need for update and 2 new sections were suggested based on median voting. However, these additions were not presented in the paper. | Ranked list of 15 guideline sections |

*(Continued)*

**Table 6.** (Continued)

| Study | Initial list of priorities | Process (steps starting with initial list and ending with final list of priorities) | Output (final list of priorities) |
|---|---|---|---|
| van der Veer, 2015 [26] | 39 topics in 4 categories | → 39 topics were drafted as a result of literature review and input from expert group<br>→ In survey round 1, participants (patients and clinicians) ranked topics and suggested 3 new ones (42 topics)<br>→ In survey round 2, participants ranked the 42 topics which were listed based on mean (standard deviation) ratings resulting in two ranked top 10 lists (for patients and clinicians) | Ranked list of 42 topics (with two ranked lists) |

medium, or low) to reflect the relative need for updating [14, 15]. One study reported a median of 167 days for the time taken to implement the prioritization process (range 18–358 days) [15].

In terms of outputs, two studies provided ranked lists of priorities [14, 26]. One study prioritized 8 out of 151 guideline documents for updating [15], another study prioritized 15 out of 35 guideline sections [14]. The third study generated a list of 42 topics from an initial list of 39 topics to be covered by the updated guideline [26].

## Discussion

### Summary of findings

We systematically reviewed the literature for prioritization exercises that have been conducted for the de novo development, update or adaptation of health practice guidelines. We identified twelve eligible studies that focused on prioritizing clinical topics and were predominantly conducted for the de novo development of guidelines; none addressed adaptation. The priority setting exercises consisted of several steps that we grouped in three phases: pre-prioritization, prioritization and post-prioritization. The two most commonly used steps were the generation of an initial list of topics (mostly by seeking input from stakeholders or by reviewing the literature) and ranking of priorities. The two least used steps were research gap analysis and having a revision mechanism. Most of the included studies used prioritization criteria as part of the exercises, with the most common criteria being the health burden and potential impact of the intervention on health outcomes. All studies involved stakeholders, particularly healthcare providers, in prioritizing guideline topics. Stakeholders were mainly involved in the generation of initial list of topics, use of criteria and ranking of priorities.

### Interpretation of findings

We observed that the generated priority topics were generally broad and non-specific. This might have been due to the fact that the vast majority of the exercises did not describe a step of refinement of the priority topics. It is essential to refine the topics in a way that would enable an easy transition from topics into meaningful questions appropriate for guideline development [31].

In addition, one of the Institute of Medicine (IOM) standards in guideline development is establishing transparency [32]. The IOM emphasizes the need for detailed and publicly accessible guideline development processes including methods for priority setting [32, 33]. Such processes would increase the credibility and the potential uptake of the end results [34]. We found only three studies reporting on the development of ethical principles to guide the conduct of the exercises.

Prioritization should be supported by an effective dissemination plan to ensure that generated priorities inform prospective research and ultimately improve health [35]. The

dissemination of priorities to researchers and funders helps in directing research agendas to guideline topics that are most important to stakeholders [36]. Although the number of prioritization exercises has been increasing over time, very few exercises reported on dissemination or implementation strategies. Consistent with findings of earlier reviews on prioritization for health research [35, 37], only three of the included exercises mentioned dissemination.

Furthermore, our reliance on a recently developed framework of prioritization criteria allowed us to categorize 66 out of the 70 criteria in the included studies. The most commonly used criterion was 'health burden'. The majority of the remaining criteria were used by two or less exercises. For instance, although equity is one of the most frequently reported criteria in the priority setting literature [38], none of the exercises considered the equity relevance of the condition. An equity-oriented approach to priority setting is important for ensuring inclusiveness [38, 39]. While this could reflect a decision by the designers of the exercises to focus on few but relevant criteria, it could also point to the failure of these exercises to be comprehensive in their use of criteria. Indeed, Nast et al. highlighted the need for such exercises to address a wide range of explicit criteria that extend beyond disease-related factors [28].

Overall, the observed variation in the prioritization steps and criteria used in the included prioritization exercises could potentially be explained by the need to tailor the decision on how to conduct a prioritization exercise to the needs of relevant stakeholders and to the available resources, such as time and funding.

A recent systematic review highlighted the opportunity to engage diverse types of stakeholders in prioritizing guideline topics [24]. Incorporating views of various stakeholders in guideline development can potentially reduce a biased selection of topics by few groups and increase transparency [7, 31]. Moreover, considering the needs of different stakeholders may improve the uptake and usability of guidelines [40]. While all exercises included in this review involved health care providers, only four and one respectively engaged researchers and patients. None engaged the other eight types of stakeholders that we assessed. Patient involvement in priority setting for guideline development has been widely supported in the literature [41–44]. It helps direct guidelines toward questions that matter most to patients, expanding beyond the interests of researchers and clinicians [45–47]. However, potential barriers to patient involvement include limited resources (e.g., lack of funds and stakeholder time), slowed down and longer process, and difficulty in identifying appropriate representatives [42, 43, 48]. In addition, guidance on how to engage patients is limited [43]. Despite potential challenges, some of the available methods for engaging patients have been evaluated, and thus can be used to ensure appropriate patient involvement [49, 50]. Moreover, maintaining regular communication with patients or their representatives facilitates meaningful engagement [51].

The online approach to engaging stakeholders was adopted by all studies. Online platforms are considered practical and cost-efficient ways of engaging stakeholders [52]. Other methods that were not frequently used in the prioritization exercises include in-person meetings and workshops. Although not widely used (for practical and financial reasons), those methods might improve interactions and discussions between stakeholders and in turn generate different priorities. In addition, face-to-face meetings are one of the knowledge exchange methods with the greatest impact on policymaking [53]. Furthermore, most stakeholders were engaged through the Delphi survey method, which is a simple consensus tool for obtaining the views of a large group of relevant stakeholders [54] using structured and iterative group interactions [55]. The Delphi method is commonly used in both guideline development and in health research prioritization [56–58], explaining its use in prioritization for guideline development.

The included studies on prioritization for updating conducted the exercises at different time points of the updating process. One exercise was implemented to identify the clinical

guidelines in greatest need of update after a surveillance process, while the other two exercises aimed to identify the topics or sections in greatest need of update for a selected guideline.

## Strengths and limitations

We used a rigorous and transparent process including a comprehensive search strategy, duplicate and independent selection, and duplicate and independent data extraction [59]. In addition, and by drawing on an extensive body of literature since the 1990s and up to July 2019, this review synthesizes almost three decades of published research on prioritization for guideline development. On the other hand, we built on two recent systematic reviews of prioritization approaches to develop our data extraction and analysis framework (e.g., how to categorize the steps of prioritization, prioritization criteria).

There are limitations to our scoping review process. First, we did not appraise the quality of the included studies. However, this is consistent with the scoping review methodology [21] and no tool has been developed for the critical appraisal of priority setting exercises. Second, we did not search the grey literature, particularly websites of guideline developing organizations, due to time and resource constraints.

## Comparison to other reviews

Our work adds to former reviews on the topic, e.g., the review by Garcia et al. which focused on the update of health decision-making tools, one of which was guidelines [20]. Consistent with our findings, Garcia et al. reported variability in the methods used to implement the prioritization exercises for updating. On the other hand, our study presents a more in-depth analysis of relevant characteristics such as the steps and criteria for prioritization exercises. Our list of criteria is consistent with, but a bit more comprehensive than the list by Garcia et al.

## Implications for practice

Our findings can assist clinicians, researchers, funders, policymakers, and other stakeholders seeking to develop health practice guidelines in prioritizing topics to be addressed. Given that there are no standard prioritization best practices for guideline development [7], it might be challenging to provide specific guidance on which prioritization exercise to use. However, the decision on whether and how to conduct a prioritization exercise should be tailored to the needs of relevant stakeholders and to the available resources, including time and funding. Furthermore, the detailed lists of identified steps and criteria can serve as a menu of options for guideline developers to select from, as judged appropriate to the context, and through a transparent decision-making process.

## Implications for future research

There is a need to develop methods and guidance for prioritization of not only topics, but also for questions and outcomes in guidelines projects. Exploring the same question of this study through the analysis of guideline handbooks would be helpful for that purpose. Further rigorous evaluation research can help with a better understanding of potential facilitators and barriers to prioritization. Moreover, and because all of the included conducted exercises were developed by researchers from high-income countries, future studies can focus on the effectiveness of the exercises in low- and middle-income countries. It is also essential to evaluate the impact of those exercises on resource allocation and on clinical outcomes.

## Conclusions

This review identified 12 prioritization exercises that addressed different aspects of priority setting for guideline development and update. The detailed lists of prioritization steps and criteria can serve as a menu of options for guideline developers to select from, as judged appropriate to the context. This review also provided insight into the types of stakeholders involved in the prioritization of health practice guidelines. Engaging diverse stakeholders, particularly patients and their representatives, is essential to align guideline development with the needs and priorities of relevant stakeholders. However, the roles of stakeholders in the prioritization processes need to be further investigated.

## Supporting information

**S1 File. PRISMA-ScR checklist.**
(DOCX)

**S2 File. Study protocol.**
(DOCX)

**S3 File. Search strategy.**
(DOCX)

**S4 File. Data extraction variables.**
(DOCX)

**S5 File. Prioritization criteria framework.**
(DOCX)

## Acknowledgments

We thank Ms. Aida Farha for her assistance in developing the search strategy.

## Author Contributions

**Conceptualization:** Amena El-Harakeh, Elie A. Akl.

**Data curation:** Amena El-Harakeh, Tamara Lotfi, Ali Ahmad.

**Formal analysis:** Amena El-Harakeh, Elie A. Akl.

**Methodology:** Amena El-Harakeh, Tamara Lotfi, Ali Ahmad, Rami Z. Morsi, Racha Fadlallah, Lama Bou-Karroum, Elie A. Akl.

**Project administration:** Amena El-Harakeh.

**Supervision:** Elie A. Akl.

**Writing – original draft:** Amena El-Harakeh, Elie A. Akl.

**Writing – review & editing:** Amena El-Harakeh, Tamara Lotfi, Ali Ahmad, Rami Z. Morsi, Racha Fadlallah, Lama Bou-Karroum, Elie A. Akl.

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
