## [Decision Letter · Decision Letter 0]

22 Nov 2019

PONE-D-19-24700

The implementation of prioritization exercises in the development and update of health practice guidelines: a systematic review

PLOS ONE

Dear Elie A. Akl 

Thank you for submitting your manuscript to PLOS ONE. After careful consideration, we feel that it has merit but does not fully meet PLOS ONE’s publication criteria as it currently stands. Therefore, we invite you to submit a revised version of the manuscript that addresses the points raised during the review process.

Please respond to the reviewers feedback and, in particular, kindly address the suggestions provided on the discussion.

We would appreciate receiving your revised manuscript by 23 December 2019. To enhance the reproducibility of your results, we recommend that if applicable you deposit your laboratory protocols in protocols.io, where a protocol can be assigned its own identifier (DOI) such that it can be cited independently in the future. For instructions see: http://journals.plos.org/plosone/s/submission-guidelines#loc-laboratory-protocols

We look forward to receiving your revised manuscript.

Kind regards,

Vicki Jane Flenady

Academic Editor

PLOS ONE

Journal Requirements:

1. As your study does not include an assessment of methodological quality and risk of bias of the included studies, please consider whether a "scoping review" or "evidence mapping" would be a more appropriate descriptor than "systematic review." Please update the title and text to reflect this.

Reviewers' comments:

Reviewer's Responses to Questions

**Comments to the Author**

1. Is the manuscript technically sound, and do the data support the conclusions?

Reviewer #1: Yes

Reviewer #2: Yes

2. Has the statistical analysis been performed appropriately and rigorously? 

Reviewer #1: N/A

Reviewer #2: Yes

3. Have the authors made all data underlying the findings in their manuscript fully available?

Reviewer #1: Yes

Reviewer #2: Yes

4. Is the manuscript presented in an intelligible fashion and written in standard English?

Reviewer #1: Yes

Reviewer #2: Yes

5. Review Comments to the Author

Reviewer #1: Summary of the research and overall impression

In this manuscript, the authors perform a comprehensive systematic review of studies describing priority setting for guideline development and update, to better understand methodologies in use for this decision-making process. This is a well-constructed, detailed and considered review. Strengths and limitations are well described, and data (including supplementary files) is presented in a clear and detailed manner. The reported findings from 12 eligible studies that there is substantial variation in the methods used to undertake prioritisations exercises whilst not surprising, provides valuable evidence to highlight the importance of explicit and transparent prioritisation processes in this area.

Major issues

None

Minor Issues

1. I am uncertain of the appropriateness of referencing a review which has been submitted but not yet published. Has your previous review on prioritization for evidence synthesis now been published, if so, please reference accordingly? (Reference number 22- Fadlallah R, El-Harakeh A, Bou-Karroum L, Lotfi T, El-Jardali F, Hishi L, et al. Prioritizing topics or questions for evidence syntheses in health: a systematic review. Manuscript submitted for publication. 2019.)

Reviewer #2: This study sought to systematically review the use of prioritisation processes for developing, updating, or adapting clinical practice guidelines in health care. Twelve eligible studies are included and described, of which all focussed on prioritising broad clinical topics. There was variation in the prioritisation exercises on various domains, including stakeholder involvement, steps of prioritisation addressed, and criteria used to prioritise topics.

This is a comprehensive and rigorous review with important implications for future research and guideline development. The rationale, objectives, and potential value of the review are clearly described. In general, the manuscript is very well-written, particularly the introduction, methods, and results. The manuscript and supplementary files are well-organised and appropriately detailed. The authors should be commended on their clear and comprehensive reporting of methods and results. Overall, this review offers several important observations about prioritisation processes, including potential deficiencies, which can assist to improve and standardise such initiatives for researchers and guideline developers.

I have a few comments and suggestions, which focus mainly on the Discussion.

Main comments:

- I felt the 'summary of findings' had omitted some interesting and important results of the review that would be worth briefly summarising in that section of the manuscript. Specifically, the dominant criteria used to prioritise topics (i.e. health burden and impact on health outcomes) and stakeholder involvement (i.e. this this was largely focussed in care providers). The latter is particularly important to mention in the summary given (diverse) stakeholder involvement is critical for maximising relevance and uptake of guidelines, as stated in the introduction and later in the discussion.

- It was striking that only one of the prioritisation exercises engaged patients or their representatives. This is a clear gap in the reviewed prioritisation exercises and it is worth exploring in the discussion the potential reasons why this might be the case. For example, it may be that research compliance issues and/or funding and resource limitations makes patient-engagement more difficult? These sorts of potential barriers (and their possible solutions) could be mentioned in paragraph 3 of the interpretation of findings (p. 30).

- Paragraph 4 of the interpretation of results (p. 30) could be expanded. It is noted that all studies adopted an online approach to engagement, likely for practical and financial reasons. It is worth mentioning the other methodologies available for priority-setting exercises, that have been potentially under-utilised (e.g. workshops and face-to-face meetings). Such methodologies may be more accessible for certain stakeholder types, and may also yield different priorities…

- As a whole, the interpretation of results felt somewhat brief. I felt it could be improved by drawing out a few more of the review's salient findings, for discussion and critical appraisal. For example, only 3 studies reported on the development of ethical principles to guide the conduct of the exercise; only 3 reported a plan for dissemination; and none included equity relevance of the condition. Do the authors see these as important gaps in current practice that should be addressed in future priority setting exercises?

- The authors state that having not searched the grey literature was a limitation of their review. It was not clear to me what was meant by "Nonetheless, these would require a different search strategy and would not reflect real life exercises.". Was the intention to say that prioritisation exercises would not be published in grey literature? Please clarify in the manuscript.

- I think the findings of this review will also be of benefit to researchers who may design and conduct priority setting activities in future. It is worth adding 'researchers' explicitly to the list given in the opening paragraph of the 'implications for practice' (p. 32).

- As the authors state under the implications for practice, "the decision on whether and how to conduct a prioritization exercise should be tailored to the needs of relevant stakeholders and to the available resources, including time and funding" (p. 32). Indeed, this may explain some of the variation in the prioritisation exercises reviewed. It is worth stating this in the discussion.

- I think it is important to (re)state in the conclusion the importance of diverse stakeholder involvement, particularly with regard to patients and their representatives, which the review has highlighted as a significant gap.

Minor comments:

- The % symbol is missing from the data label row in Table 4.

- There is an 's' missing from 'stakeholder' in the second paragraph under Stakeholder involvement (p. 20).

6. PLOS authors have the option to publish the peer review history of their article (what does this mean?). If published, this will include your full peer review and any attached files.

Reviewer #1: No

Reviewer #2: No

---

## [Author Response · Author response to Decision Letter 0]

14 Jan 2020

14 January 2020

Professor Vicki Jane Flenady

Re: “The implementation of prioritization exercises in the development and update of health practice guidelines: a scoping review” 

Dear Dr. Flenady,

Thank you for the opportunity to revise and resubmit our above titled manuscript to PLOS ONE. We thank you and the reviewers for taking the time to review the manuscript. We found the comments and suggestions very constructive and used them to improve it.

Kindly find below a point-by-point response to the comments and a description of the changes made to the manuscript.

Thank you so much and we look forward to the outcome of the peer review process.

Sincerely,

Elie A. Akl, MD, MPH, PhD

AUBMC, Department of Internal Medicine

P.O. Box: 11-0236

Riad-El-Solh Beirut 1107 2020

Beirut – Lebanon

Phone: 00961 1 374374

ea32@aub.edu.lb

The reviewers’ comments and our response are listed below.

Editor: As your study does not include an assessment of methodological quality and risk of bias of the included studies, please consider whether a "scoping review" or "evidence mapping" would be a more appropriate descriptor than "systematic review."

Please update the title and text to reflect this.

Response:

Thank you. After careful revision, we changed the study descriptor to ‘scoping review’ and updated the title and text of the manuscript accordingly.

Reviewer 1: Summary of the research and overall impression

In this manuscript, the authors perform a comprehensive systematic review of studies describing priority setting for guideline development and update, to better understand methodologies in use for this decision-making process. This is a well-constructed, detailed and considered review. Strengths and limitations are well described, and data (including supplementary files) is presented in a clear and detailed manner. The reported findings from 12 eligible studies that there is substantial variation in the methods used to undertake prioritisations exercises whilst not surprising, provides valuable evidence to highlight the importance of explicit and transparent prioritisation processes in this area.

Major issues

None

Response:

We thank the Reviewer for this very positive feedback.

Minor Issues

1. I am uncertain of the appropriateness of referencing a review which has been submitted but not yet published. Has your previous review on prioritization for evidence synthesis now been published, if so, please reference accordingly? (Reference number 22- Fadlallah R, El-Harakeh A, Bou-Karroum L, Lotfi T, El-Jardali F, Hishi L, et al. Prioritizing topics or questions for evidence syntheses in health: a systematic review. Manuscript submitted for publication. 2019.)

Response:

Thank you for this question. The paper was accepted for publication on December 11, 2019, which allows us to include a definitive citation.

Fadlallah R, El-Harakeh A, Bou-Karroum L, Lotfi T, El-Jardali F, Hishi L, et al. A common framework of steps and criteria for prioritizing topics for evidence syntheses: a systematic review. Manuscript submitted for publication. 2019. Journal of Clinical Epidemiology. 2019.

Reviewer #2:

This study sought to systematically review the use of prioritisation processes for developing, updating, or adapting clinical practice guidelines in health care. Twelve eligible studies are included and described, of which all focused on prioritising broad clinical topics. There was variation in the prioritisation exercises on various domains, including stakeholder involvement, steps of prioritisation addressed, and criteria used to prioritise topics. 

This is a comprehensive and rigorous review with important implications for future research and guideline development. The rationale, objectives, and potential value of the review are clearly described. In general, the manuscript is very well-written, particularly the introduction, methods, and results. The manuscript and supplementary files are well-organised and appropriately detailed. The authors should be commended on their clear and comprehensive reporting of methods and results. Overall, this review offers several important observations about prioritisation processes, including potential deficiencies, which can assist to improve and standardise such initiatives for researchers and guideline developers.

Response:

We sincerely thank the Reviewer for the positive and constructive assessment of our manuscript.

I have a few comments and suggestions, which focus mainly on the Discussion. 

Main comments:

Comment 1:

I felt the 'summary of findings' had omitted some interesting and important results of the review that would be worth briefly summarising in that section of the manuscript. Specifically, the dominant criteria used to prioritise topics (i.e. health burden and impact on health outcomes) and stakeholder involvement (i.e. this was largely focused in care providers). The latter is particularly important to mention in the summary given (diverse) stakeholder involvement is critical for maximizing relevance and uptake of guidelines, as stated in the introduction and later in the discussion.

Response:

Thank you for highlighting this issue. We agree and have amended the ‘summary of findings’ section by adding the following (p. 29): 

“The two most commonly used steps were the generation of an initial list of topics (mostly by seeking input from stakeholders or by reviewing the literature) and ranking of priorities. The two least used steps were research gap analysis and having a revision mechanism. Most of the included studies used prioritization criteria as part of the exercises, with the most common criteria being the health burden and potential impact of the intervention on health outcomes. All studies involved stakeholders, particularly healthcare providers, in prioritizing guideline topics. Stakeholders were mainly involved in the generation of initial list of topics, use of criteria and ranking of priorities.”

Comment 2:

It was striking that only one of the prioritisation exercises engaged patients or their representatives. This is a clear gap in the reviewed prioritisation exercises, and it is worth exploring in the discussion the potential reasons why this might be the case. For example, it may be that research compliance issues and/or funding and resource limitations makes patient engagement more difficult? These sorts of potential barriers (and their possible solutions) could be mentioned in paragraph 3 of the interpretation of findings (p. 30).

Response:

We thank the reviewer for bringing this to our attention. Indeed, the involvement of patients in only one exercise was surprising. We now reflect on patient engagement in the discussion section and assess potential reasons for not involving patients in most prioritization exercises, as per the below (p. 30):

Patient involvement in priority setting for guideline development has been widely supported in the literature [41-44]. It helps direct guidelines toward questions that matter most to patients, expanding beyond the interests of researchers and clinicians [45-47]. However, potential barriers to patient involvement include limited resources (e.g., lack of funds and stakeholder time), slowed down and longer process, and difficulty in identifying appropriate representatives [42, 43, 48]. In addition, guidance on how to engage patients is limited [43]. Despite potential challenges, some of the available methods for engaging patients have been evaluated, and thus can be used to ensure appropriate patient involvement [49, 50]. Moreover, maintaining regular communication with patients or their representatives facilitates meaningful engagement [51].

Comment 3:

Paragraph 4 of the interpretation of results (p. 30) could be expanded. It is noted that all studies adopted an online approach to engagement, likely for practical and financial reasons. It is worth mentioning the other methodologies available for priority-setting exercises, that have been potentially under-utilised (e.g. workshops and face-to-face meetings). Such methodologies may be more accessible for certain stakeholder types, and may also yield different priorities…

Response:

We agree with the reviewer and have expanded paragraph four of the ‘interpretation of results’ to examine the value of other methods (workshops and face-to-face meetings). Please see below:

Other methods that were not frequently used in the prioritization exercises include in-person meetings and workshops. Although not widely used (for practical and financial reasons), those methods might improve interactions and discussions between stakeholders and in turn generate different priorities. In addition, face-to-face meetings are one of the knowledge exchange methods with the greatest impact on policymaking [53].

Comment 4:

As a whole, the interpretation of results felt somewhat brief. I felt it could be improved by drawing out a few more of the review’s salient findings, for discussion and critical appraisal. For example, only 3 studies reported on the development of ethical principles to guide the conduct of the exercise; only 3 reported a plan for dissemination; and none included equity relevance of the condition. Do the authors see these as important gaps in current practice that should be addressed in future priority setting exercises?

Response:

Thank you, this has been detailed as follow (p. 30):

In addition, one of the Institute of Medicine (IOM) standards in guideline development is establishing transparency [32]. The IOM emphasizes the need for detailed and publicly accessible guideline development processes including methods for priority setting [32, 33]. Such processes would increase the credibility and the potential uptake of the end results [34]. We found only three studies reporting on the development of ethical principles to guide the conduct of the exercises.

Prioritization should be supported by an effective dissemination plan to ensure that generated priorities inform prospective research and ultimately improve health [35]. The dissemination of priorities to researchers and funders helps in directing research agendas to guideline topics that are most important to stakeholders [36]. Although the number of prioritization exercises has been increasing over time, very few exercises report on dissemination or implementation strategies. Consistent with findings of earlier reviews on prioritization for health research [35, 37], only three of the included exercises mentioned dissemination.

Furthermore, our reliance on a recently developed framework of prioritization criteria allowed us to categorize 66 out of the 70 criteria in the included studies. The most commonly used criterion was ‘health burden’. The majority of the remaining criteria were used by two or less exercises. For instance, although equity is one of the most frequently reported criteria in the priority setting literature [38], none of the exercises considered the equity relevance of the condition. An equity-oriented approach to priority setting is important for ensuring inclusiveness [38, 39]. While this could reflect a decision by the designers of the exercises to focus on few but relevant criteria, it could also point to the failure of these exercises to be comprehensive in their use of criteria.

Comment 5:

The authors state that having not searched the grey literature was a limitation of their review. It was not clear to me what was meant by "Nonetheless, these would require a different search strategy and would not reflect real life exercises.". Was the intention to say that prioritisation exercises would not be published in grey literature? Please clarify in the manuscript.

Response:

Thank you for the comment. Obviously, we were not as clear as we should have been. Our assessment is that unpublished priority setting exercises could be available on websites of organizations that produce guidelines. We felt that the identification of all of these organizations and sifting through their websites would be a very time and resource consuming task that might not return much information. We have clarified this point under the ‘strengths and limitations’ section as follows:

“Second, we did not search the grey literature, particularly websites of guideline developing organizations, due to time and resource constraints.

Comment 6:

I think the findings of this review will also be of benefit to researchers who may design and conduct priority setting activities in future. It is worth adding 'researchers' explicitly to the list given in the opening paragraph of the 'implications for practice' (p. 32).

Response:

We agree that researchers can benefit from the findings of our review in designing prioritization exercises for guidelines. We have made the suggested edit as follows: 

Our findings can assist clinicians, researchers, funders, policymakers, and other stakeholders seeking to develop health practice guidelines in prioritizing topics to be addressed.”

Comment 7:

As the authors state under the implications for practice, "the decision on whether and how to conduct a prioritization exercise should be tailored to the needs of relevant stakeholders and to the available resources, including time and funding" (p. 32). Indeed, this may explain some of the variation in the prioritisation exercises reviewed. It is worth stating this in the

discussion.

Response:

Thank you. This has been clarified in the discussion section as follows (p. 30):

“The observed variation in steps and criteria used the included prioritization exercises could potentially be explained by the need to tailor the decision on how to conduct a prioritization exercise to the needs of relevant stakeholders and to the available resources, such as time and funding.”

Comment 8:

I think it is important to (re)state in the conclusion the importance of diverse stakeholder involvement, particularly with regard to patients and their representatives, which the review has highlighted as a significant gap.

Response:

Thank you for bringing this to our attention. We have edited the conclusion as follows (p. 33):

“This review also provided insight into the types of stakeholders involved in the prioritization of health practice guidelines. Engaging diverse stakeholders, particularly patients and their representatives, is essential to align guideline development with the needs and priorities of relevant stakeholders.”

Minor comments:

- The % symbol is missing from the data label row in Table 4.

- There is an 's' missing from 'stakeholder' in the second paragraph under Stakeholder involvement (p. 20).

Response:

Thank you for highlighting this issue. We modified Table 4 as suggested. We also changed 'stakeholder' to ‘stakeholders’ in the second paragraph under the ‘stakeholder involvement’ section (p. 20).

---

## [Editor Report · Decision Letter 1]

4 Feb 2020

The implementation of prioritization exercises in the development and update of health practice guidelines: a scoping review

PONE-D-19-24700R1

Dear Elie Aki 

We are pleased to inform you that your manuscript has been judged scientifically suitable for publication and will be formally accepted for publication once it complies with all outstanding technical requirements.

With kind regards,

Vicki Jane Flenady

Academic Editor

PLOS ONE
---

## [Editor Report · Acceptance letter]

7 Feb 2020

PONE-D-19-24700R1 

The implementation of prioritization exercises in the development and update of health practice guidelines: a scoping review 

Dear Dr. Akl:

I am pleased to inform you that your manuscript has been deemed suitable for publication in PLOS ONE. Congratulations! Your manuscript is now with our production department. 

With kind regards,

on behalf of

Dr. Vicki Jane Flenady 

Academic Editor

PLOS ONE